# Estrogenic Effect of Probiotics on Anxiety and Depression: A Narrative Review

**DOI:** 10.3390/ijms26209948

**Published:** 2025-10-13

**Authors:** Gilberto Uriel Rosas-Sánchez, León Jesús Germán-Ponciano, Juan Francisco Rodríguez-Landa, Herlinda Bonilla-Jaime, Ofelia Limón-Morales, Rosa Isela García-Ríos, José Luis Muñoz-Carrillo, Oscar Gutiérrez-Coronado, Paola Trinidad Villalobos-Gutiérrez, César Soria-Fregozo

**Affiliations:** 1Programa de Estancias Posdoctorales por México, SECIHTI, Centro Universitario de Los Lagos, Universidad de Guadalajara, Lagos de Moreno 47460, Jalisco, Mexico; mcbjlmc@gmail.com; 2Departamento de Ciencias de la Tierra y de la Vida, Centro Universitario de Los Lagos, Universidad de Guadalajara, Lagos de Moreno 47460, Jalisco, Mexico; rosaisela.garcia@academicos.udg.mx; 3Laboratorio de Neurofarmacología, Instituto de Neuroetología, Universidad Veracruzana, Xalapa 91190, Veracruz, Mexico; lgerman@uv.mx (L.J.G.-P.); juarodriguez@uv.mx (J.F.R.-L.); 4Laboratorio de Farmacología Conductual, Department Biología de la Reproducción, D.C.B.S. Universidad Autónoma Metropolitana Iztapalapa, Iztapalapa, Ciudad de México 09340, Mexico; bjh@xanum.uam.mx (H.B.-J.); ofelia.limon@yahoo.com (O.L.-M.); 5Laboratorio de Inmunología, Centro Universitario de los Lagos, Universidad de Guadalajara, Lagos de Moreno 47460, Jalisco, Mexico; oscar.gcoronado@academicos.udg.mx (O.G.-C.); paola.villalobos2452@academicos.udg.mx (P.T.V.-G.)

**Keywords:** estrogen, *Bifidobacterium*, *Lactobacillus*, MGB axis, anxiety, depression

## Abstract

Anxiety and depression are mental disorders with significant global impact, and are especially prominent in women during times of hormonal fluctuations. The microbiota–gut–brain axis (MGB axis) has emerged as a crucial pathway in the pathogenesis of these disorders, as it directly influences the production of neurotransmitters such as serotonin (5-HT), gamma-aminobutyric acid (GABA) and dopamine (DA). In addition, they have shown estrogenic effects through enzymes such as β-glucuronidase, which modulate hormone metabolism and consequently mood. A comprehensive search of recent preclinical studies has found that probiotic intake in female rats led to significant improvements in anxiety- and depression-related behaviors. Similarly, clinical trials in certain populations, particularly women with hormonal imbalances during menopause or premenstrual syndrome, have shown promising results. However, there are still significant problems, such as the individual variability of responses and the need for controlled long-term studies. The development of specific probiotics for hormonal modulation and the implementation of personalized approaches integrating omics and neuroimaging technologies to optimize therapeutic interventions in the field of mental health are promising. Accordingly, a comprehensive search was conducted in scientific databases such as PubMed, ScienceDirect, Scopus and Web of Science. Preclinical studies investigating the estrogenic effects of different probiotic strains in animal models and in controlled clinical trials during chronic treatment were selected, excluding those studies that did not provide access to full text.

## 1. Introduction

Anxiety and depression are mood disorders that affect around 350 million people worldwide [1]. Alarmingly, these disorders have a high prevalence in women and are up to twice as common as in men, particularly during times of hormonal fluctuations such as menopause, the menstrual cycle and pregnancy [2,3]. This pronounced sexual dimorphism suggests a fundamental role for estrogens in regulating mood. As they have been shown to exert neuroprotective effects in the brain, they modulate the activity of neurotransmitters such as 5-HT, DA and GABA and promote synaptic plasticity and neurogenesis [4,5]. In this sense, estrogen fluctuations, particularly during the menopausal transition, are associated with an increase in the incidence of anxiety and depression [6].

On the other hand, the MGB axis has emerged as a new paradigm for understanding the pathophysiology of mental disorders. The bidirectional communication between the gastrointestinal tract and the central nervous system (CNS) is mediated by pathways involving the vagus nerve, the immune system, microbial metabolites and hormones [7,8]. The intestinal microbiota directly influences the production and regulation of neurotransmitters as well as the modulation of the systemic inflammatory response through modulation of the hypothalamic–pituitary–adrenal (HPA) axis, factors that significantly influence mood [9]. In recent years, probiotics have shown promise as a therapeutic intervention for anxiety and depression. Several studies have shown that probiotics improve anxiety- and depression-related behaviors without significant adverse effects in preclinical and clinical trials, demonstrating their potential as an effective treatment alternative to avoid the adverse effects and increasing costs of conventional antidepressants [10,11].

In this context, recent findings indicate that certain probiotics can have estrogenic effects by modulating estrobolome thanks to the ability of microbial genes to metabolize estrogens. By modifying microbiota, probiotics improve some menopausal symptoms by regulating estrogen metabolism and cellular signaling of these hormones [12]. Enzymes such as β-glucuronidase are involved in this mechanism, which facilitates the deconjugation and reabsorption of estrogens, thereby increasing systemic estrogen activity [13,14]. In addition, certain strains, such as *Bifidobacterium* and *Lactobacillus,* have been reported to exhibit anxiolytic and antidepressant effects in animal models, supporting the therapeutic potential of these probiotics [15]. The proposed mechanisms for the anxiolytic and antidepressant effects of these probiotics include a reduction in systemic inflammation, modulation of the immune system and the production of neuroactive metabolites such as neurotransmitter precursors, 5-HT, GABA, DA and short-chain fatty acids (SCFAs), which affect brain function [16,17,18].

However, despite promising progress, there are still opportunities for improvement, e.g., the heterogeneity of study designs, the wide variety of probiotic strains used and the population studied, which have led to inconsistent results. Furthermore, the understanding of the mechanisms involved in the estrogenic effect of probiotics and their direct effects on anxiety and depression is still unclear. Therefore, the aim of this review is to examine the literature on the estrogenic effects of probiotics on anxiety and depression and to provide a comprehensive summary of the current evidence so that researchers and healthcare professionals can stay informed of the latest findings and treatment strategies in this emerging field of integrative medicine.

## 2. Method

### 2.1. Design

This study was designed as a narrative literature review on the estrogenic effects of probiotics on anxiety and depression, analyzing information from scientific papers, book chapters, books, and official websites to provide an informative, critical, and useful synthesis of the topic. In a narrative review, there are no predetermined research questions or a specific search strategy, only a topic of interest. Recognizing that narrative reviews lack systematic methods for identifying, evaluating and synthesizing information, which could lead authors to include or exclude information to support a particular position [19,20], we describe the parameters that were used to include or exclude studies to ensure the objective inclusion of information. These included conducting a search and identifying keywords, reviewing abstracts and articles, and documenting results [21].

### 2.2. Criteria

The inclusion criteria focused on research articles, reviews and statements in which the effects of various probiotic strains were investigated in preclinical and clinical studies during chronic treatment and their mechanism of action in English-language articles. The decision to include only studies with chronic treatment was made because the positive effects of probiotics need time to consolidate and manifest clinically. Probiotics need to colonize the gut, modulate the existing microbiota and exert their immunomodulatory and metabolic effects, processes that do not occur with acute or short-term administration. Furthermore, most conditions for which probiotics are studied (e.g., irritable bowel syndrome, inflammatory bowel disease or improvement of immune function, emotional and affective disorders) are chronic and require long-term intervention. Exclusion criteria were studies without full-text access, unofficial websites, duplicate publications and doctoral dissertations.

### 2.3. Article Research

Data on the topic described in the inclusion criteria were searched in specialized databases, such as PubMed, Science Direct, Web of Science and Scopus, using a combination of specific words, such as “estrogen”, “anxiety”, “depression”, “neurobiology”, “MGB axis”, “estrogenic”, “anxiolytic”, “antidepressant”, “probiotics”, “treatment”, “animal”, “models”, “clinical trials” and “probiotic mechanism of action”.

## 3. Physiological Basis: Estrogen, Brain and Mood

Estrogens such as estrone (E1), 17β-estradiol (E2) and estriol (E3) are steroid hormones that exert effects on various physiological systems, including the CNS, in humans and experimental subjects. For years, they were considered responsible for reproductive functions, as supported by studies that have demonstrated their function in modulating brain processes and their particular role in regulating emotions and cognitive processes [22]. In this sense, estrogen receptors have been identified in brain structures such as the hippocampus, cortex and amygdala, which are involved in the regulation of emotional and affective states, suggesting a mechanism by which these hormones influence affective states [23].

### 3.1. Function of Estrogen in the Brain

Several studies have shown that estrogens play a fundamental role in neuroprotection through several mechanisms. E2 acts as a neuroprotector by reducing the effects of oxidative stress through the regulation of antioxidant enzymes such as superoxide dismutase and glutathione peroxidase [24]. This emphasizes its importance, as oxidative stress is a factor contributing to the pathogenesis of neurodegenerative and affective disorders such as anxiety and depression [25]. Other studies have reported that estrogens are able to inhibit neuronal apoptosis by activating the PI3K/Akt/Bcl-2 pathway [26]. Treatment with E2 in animal models increases the expression of the anti-apoptotic protein Bcl-2, which is able to reduce the expression of the pro-apoptotic protein Bax, thereby enhancing resistance to programmed neuronal death [27]. It has also been shown that estrogens promote neurogenesis in the hippocampus [28]. Brain-derived neurotrophic factor (BDNF) is an essential neurotrophin for neurogenesis and brain plasticity and is positively modulated by estrogens, establishing direct communication between estrogens and neurotrophic factors such as BDNF, which are responsible for neuronal protection and brain plasticity [29].

The neuroprotective effect of estrogens is also based on their anti-inflammatory effects. Estrogens have been reported to reduce the production of proinflammatory cytokines such as interleukins (IL-1β, IL-6) and tumor necrosis factor alpha (TNF-α), and also to modulate microglia activation [30,31]. This mechanism is important due to the increasing number of studies demonstrating the role of neuroinflammation in emotional and affective disorders such as anxiety and depression [32]. In addition, estrogens are able to modulate the dopaminergic, serotonergic and glutamatergic neurotransmission systems involved in the pathophysiology of anxiety and depression [33]. In the serotonergic system, for example, estrogens increase the synthesis and availability of 5-HT by upregulating the enzyme tryptophan hydroxylase, which limits 5-HT synthesis [34,35]. Additionally, they reduce the expression and function of the 5-HT transporter (SERT), which limits the reuptake of 5-HT, thereby prolonging its availability in the synaptic cleft, a similar mechanism of action to the selective serotonin reuptake inhibitors (SSRIs) [35].

Regarding the dopaminergic reward system, studies have shown that estrogens increase the density of dopamine D2 receptors in mesolimbic regions, altering the circuits associated with reward and motivation [36]. This mechanism could partly explain the susceptibility to mood disorders when estrogen fluctuations occur. In addition, E2 modulates the expression of N-metil-D-aspartate (NMDA) and alpha-amino-3-hydroxy-5-methyl-4-isoxazolepropionic acid (AMPA) receptors, promoting long-term potentiation (LTP), a fundamental process of synaptic plasticity [37]. This modulation also influences cognitive processes and emotional processing. In addition, estrogens are able to modulate the GABA-ergic system by altering the composition of GABAA receptor subunits [38] and achieving greater sensitivity to allosteric modulators such as neurosteroids like allopregnanolone [39]. This affects neuronal excitability and influences anxiety and depression at an experimental level.

Synaptic plasticity is defined as the brain’s ability to change neuronal connections in response to experience and learning. This process is influenced by estrogens, which promote structural and functional changes that facilitate information processing and neuronal adaptation [40]. At the structural level, estrogens promote dendritic spinogenesis in hippocampal neurons. E2 administration increases the density and number of spines in hippocampal CA1 neurons [41], an effect mediated by membrane estrogen receptors that activate rapid signaling cascades [42]. They also influence synaptogenesis by regulating the expression of synaptic proteins such as sinapsin I, syntaxin and postsynaptic density protein 95 (PSD-95) [43], which are essential for the maturation and formation of synapses. Other studies report that this effect is mediated by activation of the MAPK/ERK and PI3K/Akt pathways (Figure 1), culminating in cytoskeletal reorganization and the formation of new synaptic contacts [44].

### 3.2. Relationship Between Fluctuating Hormone Levels and Mood Disorders

Fluctuations in estrogen levels are closely associated with mood swings and increased susceptibility to emotional and affective disorders throughout a woman’s reproductive life [45]. This bidirectional relationship between hormones and mood highlights the complexity of the neurobiological mechanism. During the menstrual cycle, fluctuations in estrogen hormone levels are associated with mood swings [46]. In premenstrual dysphoric disorder (PMDD), for example, which affects around 3–8% of young women, affective symptoms increase during the luteal phase [47]. Recent studies suggest that it is not the low estrogen level that is the cause, but rather the speed and extent of its decline, which could trigger increased vulnerability in individuals with a predisposition to anxiety disorders or depression [48].

Functional neuroimaging studies have shown that hormonal fluctuations during the menstrual cycle modulate the activity of frontolimbic circuits associated with emotion regulation [49]. Studies have shown that the amygdala responds more strongly to negative emotional stimuli in the late luteal phase, a phase in which estrogen and progesterone levels and the metabolite allopregnanolone decrease compared to the follicular phase [50]. Other studies have found changes in the synthesis of neurotransmitter metabolites during the menstrual cycle, particularly in women with PMDD [51]. Clinical studies have observed a reduction in 5-HT metabolite concentrations during the late luteal phase in women with PMDD that correlates with greater symptom severity, suggesting an interaction between estrogen fluctuations and the serotonergic system in vulnerable individuals [52].

Menopause represents a period of vulnerability to mood disorders and increases the frequency of depressive symptoms and the occurrence of major depressive disorder (MDD) [53]. This susceptibility appears to be related to the low estrogen levels characteristic of this phase, while the lowest estrogen levels can be found postmenopause [54]. In clinical SWAN studies, a fourfold increased risk of developing depression was found in perimenopause compared to premenopause, independent of psychosocial factors and a history of MDD [53]. In this sense, the so-called “estrogenic hypothesis of depression” postulates that the reduction in estrogen levels during the menopausal transition is one of the main triggers of MDD, due to changes in the neurotransmitter system, especially in the serotonergic and noradrenergic systems, leading to the signs and symptoms characteristic of MDD [55]. In addition, neuroimaging studies have found volumetric changes in brain regions involved in emotion regulation during menopause. Other findings report a reduction in hippocampal volume associated with reduced estrogen levels, suggesting the involvement of neurostructural mechanisms in MDD associated with estrogen fluctuations [56].

The postpartum period is another risk factor for the occurrence of mood disorders. This phase is characterized by an abrupt decrease in estrogen levels after childbirth [57]. Postpartum depression affects 10–15% of women worldwide and can have significant behavioral consequences for both mother and child during development [58]. Clinical studies report that susceptibility to estrogen fluctuations can vary from woman to woman; some women may be more sensitive to hormonal changes [58]. Other studies have shown that the administration of estrogen receptor antagonists and progestogens causes similar changes to those seen in postpartum women [50]. Depressive symptoms only develop in women with a history of postpartum depression, indicating greater neurobiological susceptibility [59]. Molecular studies have shown that postpartum hormone decline alters the BDNF signaling pathway and increases levels of proinflammatory cytokines [60]. Other research has reported epigenetic patterns in genes related to estrogen response in women with postpartum depression, suggesting that epigenetic programming may regulate vulnerability during hormonal decline [61].

## 4. Role of Probiotics in Modulating the Gut–Brain Axis

The MGB axis represents a bidirectional communication pathway between the CNS and the gastrointestinal tract, which includes neural, endocrine, immunological and metabolic components [62]. Recent studies have shown that gut microbiota play a fundamental role in modulating the MGB axis, affecting behavioral and neurophysiological processes [63]. In this sense, estrogens have been shown to modulate the MGB axis and regulate mood disorders [64]. The gut microbiota has an important influence on the metabolism and availability of estrogens through enzymes such as β-glucuronidase, which cleaves estrogens and enables their absorption [65]. This subpopulation of the microbiome that metabolizes estrogens (estrobolome) determines the systemic exposure to these hormones [65]. Therefore, changes in the composition of the gut microbiota can affect estrogen metabolism. Studies have reported that antibiotic-induced dysbiosis reduces estrogen diversity, which decreases circulating estrogen levels and their metabolites, which in turn alters estrogen signaling in the brain and affects the regulation of emotions [66].

### 4.1. Influence of the Gut Microbiota on the Production and Regulation of Neurotransmitters

5-HT is a neurotransmitter involved in the regulation of mood, sleep, appetite and cognitive function [67]. Approximately 90% of 5-HT is produced by enterochromaffin cells in the gut. In this context, it has been shown that the gut microbiota can regulate 5-HT production [68], while dysbiosis leads to a decrease in serum 5-HT levels, which increases the likelihood of developing signs of anxiety and depression [68]. Normalizing the microbiota through dietary supplements such as probiotics restores 5-HT levels and thereby reduces signs of anxiety and depression in animal models. Studies in mice have shown that certain strains of probiotics are able to modulate 5-HT levels. For example, *Lactobacillus plantarum* is able to increase 5-HT levels in the hippocampus and striatum and exert anxiolytic and antidepressant effects in validated models of anxiety and depression [69]. On the other hand, *Bifidobacterium infantis* increases plasma tryptophan levels, an essential amino acid for the synthesis of 5-HT, suggesting a positive modulation of serotonergic metabolism and the regulation of mood disorders [70].

In the CNS, the most important inhibitory neurotransmitter is GABA, which is essential for the regulation of stress, social behavior and anxiety [71]. Many probiotic bacteria can produce GABA in the gut as a bioactive metabolite through the decarboxylation of glutamic acid [72]. Reports have shown that chronic administration of *Lactobacillus rhamnosus* alters the expression of GABAergic receptors in brain regions of mice, which has been correlated with a decrease in behaviors indicative of anxiety and depression [73]. This effect depends on the vagus nerve, which establishes communication between the enteric system and the brain [73]. This ability of certain probiotics to modulate the GABAergic system enhances their potential effect in modulating this neurotransmission system at the brain and gut levels. DA is a neurotransmitter that plays a fundamental role in motivation, reward, cognitive function and fine motor skills. In contrast to 5-HT, 50% of the DA present in the human body is synthesized in the gastrointestinal tract [74]. Several studies have shown that the intestinal microbiota contributes to the production and metabolism of DA and norepinephrine (NE) [75]. *Escherichia coli* and *Bacillus* spp. can increase DA and NE levels in the gut [76]. In other studies, it has been observed that mice with dysbiosis have lower DA and NE levels in plasma and intestine [77]. This effect is reversed by supplementation with probiotics such as *Lactobacillus rhamnosus*, which increases the expression of dopamine D1 receptors in the hippocampus of mice, demonstrating its effect on modulating the dopaminergic system [76].

### 4.2. Impact of Probiotics on Mental Health and Emotional Behavior

Anxiety and MDD disorders are neuropsychiatric alterations with the highest prevalence worldwide; they share common pathophysiological mechanisms such as chronic inflammation, alterations in neurotransmission systems and deregulation of the HPA axis [78]. Preclinical and clinical trials have reported positive effects of certain probiotics on anxiety and depression; in animal models, administration of *Lactobacillus rhamnosus* reduces anxiety and depressive behavior due to its effects on the expression of GABA receptors [79]. *Bifidobacterium longum* reduces anxious behavior in rodents with induced colitis, an effect mediated via the vagus nerve [80]. On the other hand, clinical studies in humans have reported that the combination of *Lactobacillus helveticus* and *Bifidobacterium longum* administered for 30 days reduces the level of anxiety and psychological distress in healthy volunteers [17]. A meta-analysis analyzed 34 randomized controlled trials that concluded that taking probiotics reduced the effects of depression symptoms but showed greater efficacy for anxiety symptoms [81]. Another meta-analysis comprising 38 studies reported that the anxiolytic and antidepressant effects of probiotics are higher in individuals clinically diagnosed with anxiety and depression than in healthy individuals [79].

The HPA axis is the most important response to stress; its dysregulation is associated with mood swings and somatic disorders. In this context, probiotics have been described as having the ability to modulate the stress response and attenuate the hyperactivity of the HPA axis [82]. The administration of a combination of *Lactobacillus farciminis*, *Lactobacillus helveticus* and *Bifidobacterium longum* prevents stress-induced hypersecretion of corticosterone, adrenaline and NE in rats, an effect associated with a reduction in intestinal permeability and decreased bacterial translocation, suggesting a specific mechanism of action [83,84]. In addition, placebo-controlled studies in humans have shown that administration of *Lactobacillus casei* Shirota for 8 weeks is able to reduce salivary cortisol levels in college students who are stressed during the exam period [85].

Changes in the gut microbiota during critical periods of gestation can have a lasting effect on the neurodevelopment of the offspring and their ability to cope with stress in adulthood [86]. Administration of *Lactobacillus reuteri* and inulin restores social deficits in mice with altered gut microbiota and also shows positive effects on cognitive functions [87]. *Bifidobacterium breve* increases the content of omega-3 fatty acids in the brain of mice, which is associated with improved performance in spatial learning and memory tests [88]. *Lactobacillus rhamnosus* and *Bifidobacterium animalis* improve recognition memory in rats, an effect that correlates with a reduction in proinflammatory cytokines in the hippocampus [89,90]. On the other hand, placebo-controlled studies in humans have shown that administration of *Bifidobacterium breve* for 12 weeks significantly improves immediate memory and sustained attention in older adults [91].

### 4.3. Potential Mechanisms of Action of Probiotics

Chronic inflammation and neuroinflammation have been linked to the development of psychiatric disorders such as anxiety, depression and neurodegenerative problems [92]. Experimentally, several probiotic strains have shown anti-inflammatory properties that contribute to their neuromodulatory effects in the brain [93]. *Lactobacillus rhamnosus* reduced the expression of proinflammatory cytokines IL-1β and TNF-α in the mouse model of colitis and also showed anxiolytic effects in the elevated plus-maze test [94,95,96]. In clinical studies, *Bifidobacterium infantis* reduced levels of proinflammatory cytokines in patients with fatigue and irritable bowel syndrome. These anti-inflammatory effects could reduce neuroinflammation and thus protect the integrity of the blood–brain barrier (BBB) and improve neuronal function in regions critical for the regulation of emotional and affective disorders [97]. Probiotics such as *Bifidobacterium* spp and *Faecalibacterium prausnitzii* are capable of producing SCFAs through bacterial fermentation of dietary fiber. The main SCFAs produced by probiotics include acetate, butyrate and propionate, which have shown anti-inflammatory effects in preclinical studies [98,99]. In this context, butyrate has been shown to cross the BBB, inhibit histone deacetylase, modulate the expression of genes indicative of inflammation and reduce proinflammatory activation of microglia, suggesting an indirect mechanism by which probiotics could modulate neuroinflammation [100].

The gut houses approximately 70% of immune cells, and it is here that a complex interaction with the intestinal microbiota takes place that promotes the development and functioning of the immune system [101]. Several studies have reported that probiotics have the unique ability to modulate innate and adaptive immunity through multiple mechanisms. For example, *Lactobacillus casei* Shirota promotes the activation of natural killer cells and the production of interferon gamma (IFN-γ) [102]. Immune signaling between the brain and the gut occurs via multiple pathways, including the vagus nerve, the circulatory system, the immune system and the gut microbiota. In this sense, probiotics are able to modulate these signaling pathways and promote an anti-inflammatory immune balance that has a positive impact on neurological and psychiatric health [103].

Several neuropsychiatric disorders have been associated with a disorder of the intestinal barrier characterized by increased permeability (“leaky gut”) [104]. Bacterial lipopolysaccharides (LPS) and other antigens can enter the bloodstream due to this increased permeability and trigger systemic inflammatory responses that impair brain function [104]. Several probiotic strains have demonstrated their ability to strengthen the intestinal barrier and prevent the crossing of molecules that could impair brain function [105]. In studies on human intestinal cell cultures, it was observed that *Lactobacillus plantarum* MB452 improved the expression of tight junction proteins (occludin, ZO-1). On the other hand, *Escherichia coli* Nissle 1917 strengthened the epithelial barrier by inducing the expression of ZO-2 and the redistribution of ZO-1 [106,107]. Additionally, *Bifidobacterium longum* inhibited the increase in intestinal permeability caused by prolonged stress in animal models and reduced depression-like behaviors. These results suggest that ingestion of probiotics can restore the integrity of the intestinal barrier and stop bacterial translocation with subsequent systemic immune activation, which is an indirect but important mechanism [108].

The literature describes that the intestinal microbiota is actively involved in the metabolism of compounds with neuroendocrine effects, such as sex hormones, glucocorticoids and thyroid hormones. Reports suggest that probiotics are able to modulate these processes by expressing enzymes such as β-glucuronidase, which deconjugates hormonal metabolites, allowing their reabsorption and prolonging their biological activity [109]. Recent studies have shown that certain probiotic strains can modulate β-glucuronidase activity, improving estrogen metabolism through the action of the estrobolome [109]. Therefore, alterations in the gut bacterial ecosystem affect circulating estrogen levels and favor the occurrence of neuropsychiatric disorders such as postpartum depression, premenopausal depression and anxiety. In this sense, probiotics are considered an interesting alternative for the treatment of these disorders due to their modulating effects on estrogen metabolism [110].

Studies have shown that estrogens can alter the composition and function of the gut microbiota. In addition, estrogen receptors have been shown to be expressed in intestinal epithelial cells, which are able to modulate the expression of antimicrobial proteins that improve the permeability of the intestinal barrier [65]. Therefore, the interaction between estrogens, gut microbiota and mood represents a new field with promising clinical implications [65]. Probiotic supplementation reduced ovariectomy-induced depressive symptoms in animal models, an effect associated with serotonergic restoration and the reduction in proinflammatory cytokines. Increased intestinal permeability allows the translocation of LPS into the systemic circulation, triggering an inflammatory response, neuroinflammation and subsequently depressive symptoms [84]. Therefore, tryptophan-derived bacterial metabolites such as indoles act as ligands for estrogen receptors, a pathway through which probiotics could modulate estrogen signaling in the brain [111]. This interaction represents a promising therapeutic target for disorders associated with estrogen fluctuations, as estrogens not only act as modulators of neurotransmission systems, but also influence neuroprotection, synaptic plasticity and gut–brain communication (Figure 2).

## 5. Mechanisms of Action of Estrogenic Effect of Probiotics

Estrogens are mainly synthetized in the ovaries, adrenal glands and adipose tissue. Two estrogens, E1 and E2, are mainly synthesized in the ovaries, while E3 is the primary metabolite of the degradation of E1 and E2 in the liver [112]. Estrogens are present in the bloodstream in two forms: conjugated and free. The former predominates, but the biologically active form in the blood is the free expression. Estrogens remain inactive in the circulation by bonding primarily to sex hormone-binding globulin (SHBG) [112]. Once deconjugated, they regain their biological activity and bind to their alpha (ERα) and beta (ERβ) estrogen receptors and membrane receptors in their target tissues. Estrogens are metabolized in the liver in a process that involves a series of reactions to form a biologically inactive conjugated form, a portion of which is excreted in the urine and bile [113]. The conjugated form excreted in bile enters the intestine where it is deconjugated by β-glucuronidase in the gut microbiota. A portion of this biologically active deconjugated form is reabsorbed through the intestinal mucosa and portal vein to be enterohepatically recycled [113].

The gut microbiota performs functions both locally and over long distances, involving metabolites and intermediary hormones [114,115]. These functions include the metabolization of estrogens [116]. The estrobolome is defined as the repertoire of genes in the intestinal microbiota that can metabolize estrogens [12]. The GUS gene of the estrobolome encodes a member of the glycoside hydrolase 2 (GH2) family, which includes β-glucuronidases, β-mannosidases and β-galactosidases [117]. Intestinal microbiota-derived β-glucuronidases and β-galactosidases are involved in estrogen metabolism in the human body [118,119]. Intestinal microbial β-glucuronidase (gmGUS) is the one that has been studied most frequently for estrogen metabolism. The human microbiome GI project identified 279 GUS genes, of which 93.5% were taxonomically assigned to *Bacteroidetes* (52%), *Firmicutes* (43%), *Verrucomicrobiota* (1.5%) and *Proteobacteria* (0–5%) [120,121]. The β-glucuronidase bacteria are distributed in the *Enterobacteriaceae* family in different gender of *Firmicutes* (*Lactobacillus, Streptococcus, Clostridium, Ruminococcus, Roseburia* and *Faecalibacterium*) and one species of *Actinobacteria* (*Bifidobacterium dentium*) [122,123,124]. In a study from a metagenomic gut focus, bacterial genomes were identified in strain Firmicutes with enzymatic β-D-glucuronidase activity. The identified genomes were from strains of *Ruminococcaceae, Lachnospiraceae* and *Clostridiaceae* [125]. By studying 40 different bacterial strains in human feces, the authors found that the *Firmicutes* bacterial groups within the *Clostridia* groups exhibited β-glucuronidase activity [124]. Based on an in vitro analysis, [126] stated for the first time that 35 GUS enzymes of the human microbiome reactivated estrone-3-glucuronide and estradiol-17-glucuronide in E1 and E2, respectively [126].

Another aspect is that the intestinal microbiota does not only regulate circulating estrogens, because estrogens can also influence the diversity and composition of the intestinal microbiome [113]. Changes in estrogen levels affect the gut microbiota in different physiological phases [127]. During menopause, serum estrogen levels decrease and significantly alter the composition and quantity of the gut microbiota [128]. A study of postmenopausal women found that they had lower diversity in their gut microbiota and lower levels of microbial β-glucuronidases—such as *Bifidobacterium animalis*– than premenopausal individuals [129,130]. In adult rats, ovariectomy (OVX) induced changes in the abundance of two principal phyla: *Bacteroidetes* and *Firmicutes* [131]. Female ApoE-/-OVX mice showed a reduction in the composition of the species of the intestinal microbiota, although after estrogen supplementation the diversity and composition of their microbiota was restored to the level of ApoE-/- mice fed HFD instead of OVX [132]. The diversity and composition of the intestinal microbiota is thus influenced by estrogen concentrations [133], while the ERβ estrogen receptor also appears to influence the composition of the microbiota [113]. In female knock-out mice tested for the ERβ estrogen receptor, the results showed that the gut microbiota were affected in a specific way when the diet administered was changed from one containing estrogenic isoflavones to one without this element. In this study, the main phyla, including *Proteobacteria, Bacteroidetes* and *Firmicutes*, differed depending on the state of the ERβ [113].

### Relation Between Estrogens and the Intestinal Microbiota in Mood Regulation

Both estrogens and the intestinal microbiota play a role in regulating the synthesis of 5-HT, a neurotransmitter involved in the regulation of mood and the development of depression [113]. 5-HT is synthesized in the digestive tract, as well as in the nervous and immune systems. The gut microbiota synthesizes this neurotransmitter, while estrogens at the neuronal level regulate its synthesis, SERT, degradation by the enzyme monoamine oxidase (MAO) and the 5-HT1A and 5-HT2A receptors. Under normal conditions, the interaction between gmGUS and estrogen maintains the bodily homeostasis of numerous physiological processes [134]. However, when this balance breaks down, as is the case with a decrease in estrogen metabolism, it leads to associated disorders, which may include menopausal syndrome, anxiety and depression, along with dysbiosis of the gut microbiome, which can exacerbate these pathological processes [135]. In addition, dysbiosis of the gut microbiota can affect estrogen levels and serotonin synthesis.

The deficiency of 5-HT in the CNS is a factor that can cause depression and anxiety. In this context, 2% of the essential amino acid tryptophan ingested with food is converted into 5-HT [136]. Of this 2%, 95% is produced by enterochromaffin cells in the mucosa—microorganisms that are part of the gut microbiota—and neurons in the nerve plexuses of the submucosa and muscle layers of the gut [137]. Bacteria such as *Lactococcus*, *Lactobacillus, Streptococcus, Escherichia coli* and *Klebsiella* can produce 5-HT through the expression of tryptophan synthetase [138]. The intestinal microbiota communicates with the brain via three main pathways: neural (vagus nerve, enteric nervous system), immune (cytokines) and endocrine via the HPA axis. Alterations in the MGB axis can lead to mental disorders, as there is evidence that the gut microbiota interacts with the host brain and may play a fundamental role in depression.

Ninety percent of all tryptophan is oxidized via the kynurenine pathway (KP). Tryptophan is not only the precursor of 5-HT synthesis, but is also metabolized to KP by the enzyme indoleamine 2,3-dioxygenase (IDO), which is found in various organs such as the brain, gastrointestinal tract and liver [138]. On the other hand, KP is metabolized into kynurenic acid (KYNA) and quinolinic acid (QUIN) [139]. KYNA is a neuroprotective antagonist of the NMDA receptor, while QUIN is a neurotoxic agonist of the NMDA receptor [140]. The KP signaling pathway is modulated by the activation of IDO in response to immunological and inflammatory effects. In patients with inflammatory bowel disease, overexpression of IDO has been found in both the colonic mucosa [141,142] and the brain [143], suggesting that overactivation of the KP signaling pathway is involved in inflammatory processes at the gastrointestinal and cerebral levels [142,144,145].

It is assumed that activation of the KP signaling pathway leads to a decrease in 5-HT at the central level and is thus involved in the etiopathogenesis of depression [146]. Abnormal KP activation has been observed in patients with depression [146] as well as in animal models of stress-induced depression, suggesting that it is capable of altering the composition of the microbiota, changing the balance of intestinal permeability and activating the KYN signaling pathway [143,147,148,149]. A study in a mouse model in which the animals were exposed to chronic stress by restriction showed depressive behaviors with a pronounced activation of the KYN pathway in the brain and intestine, especially in the colon. IDO concentrations in the brain and gut were also significantly increased compared to control mice [143].

Several studies have reported changes in the gut microbial profile of patients with MDD, particularly in *Firmicutes* (mainly the genera *Clostridium, Enterococcus, Lactobacillus* and *Faecalibacterium*) and *Bacteroidetes* (especially the species *Bacteroides* and *Prevotella*). Other phyla that have been shown to be altered in MDD patients are *Proteobacteria*, *Actinobacteria*, *Fusobacteria* and *Verrucomicrobiota* [150]. These individuals also show an increase in *Eggerthella* but a decrease in *Sutterella* [151]. Despite these findings, the mechanisms underlying depression in the MGB axis remain unclear. In recent years, the importance of dysbiosis in inflammatory states has been emphasized in the etiopathogenesis of depression, but this phenomenon remains poorly understood, although a recent study linked inflammation to dysbiosis. Observations of adolescents with depression show that these subjects had a decreased relative abundance of *Faecalibacterium*, *Roseburia*, *Collinsella*, *Blautia*, *Phascolarctobacterium*, and *Lachnospiraceae*, which was associated with disruption of the metabolic Trp-Kyn pathway [149]. In a second phase, the researchers transplanted the fecal microbiota from healthy adolescent volunteers into adolescent mice that were depressed by restriction and chronic stress. They found that the depressive behavior of the mice improved significantly, with *Roseburia* playing a crucial role, as its effective colonization in the large intestine led to a remarkable increase in 5-HT levels there and in the brain by promoting the expression of tryptophan hydroxylase-2 (TPH2) or -1 (TPH1), reducing KYN and QUIN levels in the brain and colon, and restricting the rate-limiting enzyme–IDO1—responsible for the formation of kynurenine [149].

Several other studies have pointed to a role of E2 in regulating the serotonergic system [35], mainly through the interaction with its ERβ estrogen receptors located in different areas of the brain (e.g., hippocampus and amygdala), which are important for learning, memory and emotional regulation [152,153], and in the dorsal raphe nucleus (DRN) [154], which is responsible for serotonergic projections to various brain regions, including the aforementioned hippocampus and amygdala. Various studies have shown that estrogens regulate the genic expression of TPH, the 5-HT2A receptor, SERT, monoaminoxidase A and the activity of the 5-HT1A receptor [35,155,156]. This suggests that estrogens regulate the activity of the limiting enzyme for 5-HT synthesis—TPH-2—via ERβ receptors present in neuronal cells [35,157,158] through genomic mechanisms, inducing their transcription [159] and thereby generating 5-HT synthesis.

Estrogens also affect SERT, which retrieves excess 5-HT from the synaptic space to the presynaptic terminal [35]. In addition, they regulate the density and genic expression of SERT [155,156] in different brain regions [160,161] including the DRN [162] and the ventromedial nucleus (VNM) [160]. Therefore, they positively regulate SERT expressions. However, there are some differences between these results. In female OVX mice, the binding density of SERT in the hippocampus decreased significantly compared to controls [163], while a study in OVX macaques showed a reduction in the intensity of SERT expression and the number of 5-HT-positive cells in the DRN compared to intact animals [162]. E2 also suppressed the activity of the 5-HT1A receptor activity through its G-protein-coupled membrane receptors (GPCR) [164]. The effect of E2 on the GPCR is to reduce the firing rate of 5-HT neurons [165] by uncoupling 5-HT1A receptors from their G-protein [166], thereby increasing the activity of serotonergic neurons.

Estrogens regulate the serotoninergic system and there is evidence that their deficiency is involved in the development of depression. A decrease in estrogens has been linked to inflammation and activation of IDO in the brain. In female rats and mice, OVX induces depressive behaviors, decreases 5-HT levels and causes an increase in IDO levels in IL-6, receptor type Toll-4 and NF-KB in the hippocampus [167,168]. These changes were reversed by E2 treatment, which improved depressive behavior, inflammatory factors (NF-KB, TNF-α and IL-6) in IDO1 and activation of the IDO1-mediated TRP/KYN pathway. This suggests that estrogen inhibits inflammation and maintains 5-HT levels in the hippocampus [167,168], an effect that appears to be mediated via the ERβ receptor [169]. Reduced estrogen can cause depressive disorders in women, a process in which the microbiota appears to be involved. In a study of premenopausal women with depression, *Klebsiella aerogenes* were isolated in feces, which degrade estradiol. The *Klebsiella aerogenes* were transplanted into mice using a probe, which resulted in a decrease in estradiol levels and depressive behaviors. The authors identified 3β-hydroxysteroid dehydrogenase (3β-HSD) as the factor responsible for estradiol degradation. The prevalence of *Klebsiella aerogenes* and 3β-HSD was greater in premenopausal women with depression than in subjects without depression [170]. 3β-HSD also degrades testosterone in males and was identified in *Mycobacterium neoaurum* from men with depression with testosterone deficiency [171]; however, *Pseudomonas nitroreducens* was also reported to have 3/17β-hydroxysteroid dehydrogenase (3/17β-HSD), a gene responsible for the degradation of testosterone in men [172]. Taken together, these results suggest that 3β-HSD expressed by gut microbes may be associated with depressive symptoms due to the degradation of estradiol and testosterone in women and men, respectively.

## 6. Preclinical Evidence of Use Probiotic

Recently, the link between an imbalance in the gut microbiota (gut dysbiosis) and depression has been highlighted, as dysbiosis affects the production of neurotransmitters, some neurotrophic factors, hormones and tryptophan metabolism, etc. Probiotics have demonstrated their anxiolytic and antidepressant efficacy at the preclinical level in models of the induction of metabolic changes and induced stress.

Regarding the anxiolytic properties of probiotics, [173] observed that C57BL/6 mice fed a high-fat diet exhibited a shorter latency to the first transition from the illuminated to the dark compartment during the light–dark test, indicating an increase in anxiety-associated behavior. This effect was reversed by treatment with *Lacticaseibacillus rhamnosus* LB1.5 (3.1 × 10^8^ CFU/mL) for 13 weeks, three times a week [173]. In addition, the treatment significantly increased the distance traveled and time spent in the illuminated compartment, indicating a possible anxiolytic effect, and a reduction in IL-6 was observed. In the same strain of mice, sleep deprivation and water restriction tests were shown to induce anxiogenic effects and visceral hypersensitivity mediated by hyperactivation of the HPA axis. Daily administration of *Limosilactobacillus reuteri* (strain WLR01; 2.0 × 10^9^ CFU/0.1 mL) over a 9-week period attenuated these effects and increased distance traveled (strain WLR01) and time spent in the open arms (for both strains WLR01 and WLR06) in both the elevated plus-maze and Y-maze tests. These behavioral changes were associated with a significant reduction in cortisone and corticotropin-releasing hormone (CRH) levels, suggesting a positive modulation of the HPA axis by the probiotic [60].

Administration of the mixture of *Bifdobacterium, Lactobacillus acidophilus*, *Lactobacillus rhamnosus, Bifodan* A/S (1 × 10^9^ CFU/mL, 200 µL/daily for 30 days) reverses the anxiogenic effect of chronic ethanol exposure in C57BL/6N mice, evaluated in the elevated plus-maze and open-field test, and also decreases the levels of proinflammatory proteins such as NLRP3, NF-κB, and IL-1β in the hippocampus [174]. In terms of reversing the deleterious effects of stress that lead to the development of anxious behavior, probiotics have also shown important protective effects. In a study using maternal separation and the chronic unpredictable mild stress (CUMS) test as triggers of anxiety and hopelessness, the administration of *Bacillus coagulans* Unique IS-2 (2 × 10^9^ CFU daily for 6 weeks) in Sprague Dawley rats resulted in a reduction in the percentage of entries and open-arm retention in the elevated plus-maze back to control levels [175].

In addition, it also succeeded in reversing the increase in immobility time in the forced swim test and the percentage of sugar water intake, showing important anxiolytic and antidepressant effects, respectively [175]. In addition, the probiotic was found to reverse the reduction in BDNF in the hippocampus of male rats induced by the combination of maternal separation and CUMS. This model also induces an increase in proinflammatory cytokines such as TNF-α and C-reactive protein (CRP), which negatively affect the integrity of epithelial tight junctions and promote the entry of peripheral proinflammatory mediators into the brain [63]. TNF-α increases in the frontal cortex and hippocampus and CRP in the frontal cortex of males and females, while it also increases both CRP and IL-1β in the hippocampus, although only in females. All of these neuroinflammatory effects were reversed by the administration of *Bacillus coagulans*, as was the increase in the plasma levels of L-tryptophan in female rats, L-kynurenine and the metabolite 3-hydroxyanthranilic acid in male and female rats [175].

Treatment with *Bacillus coagulans* showed sex-dependent effects on changes in monoamines in the frontal cortex of rats. In male rats, NE levels were not altered by either CUMS or the prebiotic, but in female rats there was a reduction that could not be reversed by the prebiotic. In the case of DA, CUMS increased levels regardless of sex; the probiotic reversed this effect in males, while in females it increased DA levels twice as much as the control. In the case of 5-HT, CUMS caused a significant reduction in levels, and the probiotic reversed this effect twice as effectively in females as in males. No changes in DA levels were observed in either sex [175]. It is worth noting that in the same study, despite the above changes, anxiolytic and antidepressant effects were found regardless of gender, indicating different neurobiological mechanisms related to the interaction with the gonadal hormone systems.

Regarding the antidepressant properties of probiotics, [176] found that the administration of a probiotic mixture of *Bifidobacterium breve* (25%), *Lactobacillus plantarum* (25%) and *Lactobacillus paracillus* (25 ) for 10 consecutive days led to an improvement in the antidepressant effect, *Lactobacillus paracasei* (25%) and *Lactobacillus helveticus* (25%) at concentrations of 8 × 10^9^ CFU reverses the effects on depression-like behaviors induced by chronic unpredictable stress (CUS), such as the reduction in sucrose levels and an increase in immobility time in the tail suspension and forced swim tests. In addition, probiotic treatment suppressed indicators of neuroinflammation in the hippocampus and medial prefrontal cortex, such as increases in mRNA and protein levels of IL-1β, IL-6 and TNF-α. These proinflammatory cytokines have been linked to the development of depression-like behaviors in animal models [177]. CUS leads to a decrease in levels of anti-inflammatory mediators such as IL-4, IL-10 and Ym-1, an effect that is also reversed by the probiotic treatment used by [176]. A suppression of nitroxidative stress has also been observed, as evidenced by the altered levels of biomarkers such as nitrites (NO), malondialdehyde (MDA) and reduced glutathione (GSH) in the hippocampus and medial prefrontal cortex, factors that have been linked to the development of depression-like behaviors in animal models [178], and neurotoxic effects such as the reduction in BDNF levels [179], biomarkers that were restored by probiotic treatment in mice exposed to CUS [176].

Probiotics, similar to clinically effective antidepressants such as paroxetine, have been shown to reverse the prodepressant effects induced by the CUMS model in rats in the forced swim test and the open field test. In addition to restoring the reduction in serotonin and norepinephrine levels in the hippocampus and the increase in adrenocorticotropic hormone (ACTH) and corticosterone (CORT) in plasma, a reduction in *Lactobacillus* and *Bifidobacterium* levels in the caecum of rats and an increase in *Escherichia coli* and *Escherichia faecalis* were also found as an effect of CUMS, which was reversed by both probiotics and paroxetine [180]. This supports the hypothesis that gut flora remodeling is closely associated with a reversal of depressive behaviors. Another study, in addition to finding the previously described effect of CUMS, shows that probiotics such as *Bifidobacterium longum* and *L. rhamnosus* tend to decrease 5-HT levels in the colon and increase them in the hippocampus and prefrontal cortex, which is associated with the antidepressant effect found in the forced swim test and preference for sucrose consumption [181].

Several studies suggest that probiotics from the Lactobacillus family have endocrine and neurochemical regulatory capacities that are associated with antidepressant effects. *Lactobacillus rhamnosus* JB-1 has been shown to lower corticosterone levels [15] and increase GSH, glutamate and NE transporters [182]. *Lactobacillus casei* increases NE, 5-HT, DA and BDNF levels [183]. *L. paracasei* HT6 normalizes the levels of ACTH, CORT, GR, 5-HT, NE and DA [184].

There is evidence of dysbiosis in OVX mice, which is associated with anxiety-like behavior in the light/dark test. This effect is reversed by treatment with estradiol, which reduces the expression of c-fos in the paraventricular nucleus of the hypothalamus, the medial preoptic area and the subparafascicular nucleus of the thalamus and is associated with the restoration of the gut microbiota and the anxiolytic effect observed [185]. These findings suggest the importance of investigating the effects of the decline in ovarian hormones during surgical menopause on the development of anxiety disorder. In OVX mice, anxiety-inducing effects were also found in models of the open field and elevated arm maze, as well as prodepressive behaviors in the tail suspension and forced swim test [186]. These effects are reversed by *Prevotella histicola* treatment, which additionally reverses the damage to neuronal structure and inflammation in the CA3 region of the hippocampus as well as neuronal protein synthesis induced by ovariectomy-induced estradiol deficiency. Interestingly, ovariectomy did not lead to changes in serotonergic and dopaminergic receptor levels as well as BDNF protein and mRNA levels, but *Prevotella histicola* significantly increased these levels, suggesting that this is related to the promotion of neuronal proliferation [186]. In addition, the probiotic reversed the dysbiosis caused by ovariectomy.

In view of the proven anxiolytic and antidepressant properties of probiotics, their synergistic effect with clinically effective drugs has also been investigated. The combination of a symbiotic mixture of 6.25 × 10^6^ CFU (probiotics *Lactobacillus casei*, *L. acidophilus*, *L. rhamnosus*, *L. bulgaricus*, *Bifidobacterium breve*, *B. infantis*, *Streptococcus thermophilus* + prebiotic: fructooligosaccharides) with low effective doses of the antidepressants doxepin (1 mg/kg), venlafaxine and fluvoxamine (15 mg/kg) was studied in mice subjected to the marble-burying and forced swim tests. It was found that the combination of treatments reduced compulsive behavior and immobility time after 7 and 14 days of treatment, respectively [187]. Finally, it is interesting to note that probiotics (*Lactobacillus rhamnosus*, 1 × 10^9^ CFU, for 30 days) can reverse some side effects of paroxetine in C57BL/6xBALB/c mice, such as the reduction in sperm quality and natural fertility [188], a fact closely related to testosterone levels, which are responsible for spermatogenesis; *Lactobacillus rhamnosus* has been documented to regulate testosterone levels [189]. Table 1 summarizes the role of probiotics, their effects on estrogen, and how they positively regulate anxiety and depression.

## 7. Clinical Evidence of the Effects of Probiotics on Depressive and Anxiety Symptoms in Women

According to the World Health Organization [1], anxiety and depression symptoms are more common in women than in men. In addition to this higher prevalence, women have also been found to have higher levels of severity and comorbidity. These differences have been attributed to social, economic and cultural factors; however, from a biological perspective, one of the most important explanatory factors is the influence of hormonal fluctuations, particularly estrogen levels [190]. Such hormonal fluctuations occur in different phases of a woman’s life, e.g., in the premenstrual phase, during pregnancy, after childbirth, during menopause and in pathological conditions such as polycystic ovary syndrome (PCOS). These fluctuations are associated with an increased susceptibility to mental disorders, particularly symptoms of anxiety and depression [191].

Recently, there has been growing interest in understanding the role of the gut microbiota in brain function and its impact on mental health. This has led to research into therapeutic alternatives such as probiotics that can influence the MGB axis [192]. This line of research has been extended to women who, as previously mentioned, have a higher prevalence rate of anxiety and depression. Conventional pharmacological treatments for these conditions typically include antidepressants and hormone replacement therapies, both of which are associated with significant side effects such as insomnia, decreased libido and a potentially increased risk of breast cancer [193]. The identification and evaluation of interventions with fewer adverse effects, such as probiotics, is therefore a promising but under-researched area.

One of the first phases in a woman’s life that is characterized by significant physical and psychological changes is the onset of menstruation. This period is divided into two phases—the follicular phase and the luteal phase—during which the levels of ovarian hormones such as E2 and progesterone fluctuate greatly. At the beginning of the cycle (day 1), for example, both hormones are at a low level, then rise during ovulation and fall sharply towards the end of the luteal phase if pregnancy does not occur [47]. This drop in hormones is associated with the symptoms of premenstrual syndrome (PMS), which include emotional and affective disorders such as anxiety and depression that disproportionately affect women [48].

Despite the high susceptibility to mental disorders during this phase, research on the effects of probiotics on depressive and anxiety symptoms in women is still limited. One study investigated the effects of a 10-week probiotic intervention—including *Bifidobacterium longum* subsp. *longum* OLP-01, *Lactiplantibacillus plantarum* PL-02 and *Lactococcus lactis* LY-66—in a group of 65 women aged 18 to 40 years [194]. The results indicated a significant improvement in overall premenstrual symptoms, especially in emotional areas, such as reductions in anger, depression, crying episodes, anxiety, fatigue and social limitations compared to the placebo group.

Another study investigated the effect of probiotics on anxiety and depression symptoms in women with PMS. Daily administration of *Lactobacillus gasseri* CP2305 tablets for 183 days in young women (aged 20 to 35 years) resulted in a significant reduction in anxiety, depression and fatigue. Remarkably, the emotional improvement was associated with an increase in estrogen and progesterone levels during the luteal phase. The authors suggest that these hormones may represent a new target for *Lactobacillus gasseri* CP2305 via the MGB axis. In addition, the improved gut conditions, as evidenced by reduced constipation, may have influenced the metabolism of reproductive hormones and contributed to the alleviation of emotional symptoms [195].

Similarly, pregnancy is a critical phase in the female reproductive cycle. During this time, women face significant emotional challenges related to the responsibility of pregnancy and the associated uncertainty, as well as significant hormonal changes during pregnancy [196]. These changes have been associated with the onset of psychiatric disorders such as anxiety and depression, which, if left untreated, can have a negative impact on both maternal health and fetal development [197,198]. A transient increase in cortisol levels, especially in the third trimester, and increased maternal stress associated with increased catecholamine release have been linked to a higher risk of postpartum anxiety and depression [199].

Despite these findings, there is little evidence to support the use of probiotics as a therapeutic strategy during pregnancy. However, it has been suggested that probiotics can restore gut dysbiosis, improve gastrointestinal integrity, reduce systemic inflammation, regulate endocrine activity and support neurotransmission. In addition, probiotics are considered safe for use during pregnancy [200]. In one study [200], capsules containing *Lactobacillus rhamnosus* GG and *Bifidobacterium lactis* BB12 were administered daily for 36 weeks of pregnancy. No significant improvements in depression, anxiety, functional health or general well-being were observed between baseline and 36 weeks’ gestation. The authors pointed out the uncertainty regarding whether dosing was sufficient to affect mental health and suggested that low baseline levels of mental symptoms may have led to a floor effect that limited the detection of intervention-related changes. Similarly, [201] examined the effects of *Bifidobacterium longum* NCC3001 in pregnant women in their third trimester (21 years and older) and found no significant effects on anxiety or depression symptoms. These results suggest that probiotics may not have a protective effect against anxiety or depression symptoms during pregnancy; however, the evidence is limited and inconclusive. Further studies with a greater diversity of probiotic strains are needed to clarify their potential role in the prevention of mental disorders in pregnant women.

The postpartum period is another critical phase in a woman’s life, especially for mood regulation, as the likelihood of developing postpartum depression is high [202]. A significant number of women are affected shortly after giving birth, with an estimated prevalence of 10–20%. During this phase, women experience numerous biological changes, including fluctuations in estrogen and progesterone, fluctuations in lactogenic hormones such as oxytocin and prolactin and an increase in stress hormones such as cortisol [203].

At this stage, only one study in Italy was identified in which the effects of probiotic supplementation were investigated in 200 healthy pregnant women aged 18 to 50 years. The participants were given a combination of *Limosilactobacillus reuteri* PBS072 and *Bifidobacterium breve* BB077 for 90 days. The results showed a reduction in anxiety and depression symptoms. However, the study did not investigate the possible link between these effects and hormone levels—such as estrogen and progesterone—which are directly involved in mood regulation [204]. Given the crucial importance of this stage of life for mother and child, further research is needed to clarify the effects of probiotics on anxiety and depression after childbirth.

Another phase with profound physiological and emotional effects during a woman’s reproductive life is the menopause, which is characterized primarily by a significant drop in estrogen and progesterone levels. These hormones are important precursors in the female brain and play a key role in modulating various neurotransmitter systems, such as the serotonergic, dopaminergic and GABAergic pathways, all of which are involved in mood regulation [205]. Consequently, an abrupt or progressive drop in hormones often leads to mental disorders, including anxiety and depression, especially during menopause [206]. In this regard, investigating the effects of probiotics during this reproductive phase should be a priority, particularly with regard to their potential to modulate anxiety and depressive symptoms.

Nevertheless, clinical studies investigating the effects of probiotics on the mental health of menopausal women are scarce or almost non-existent. The only identified study involved Japanese premenopausal women aged 40 to 60 years. The administration of *Lactobacillus gasseri* CP2305 tablets once daily for six menstrual cycles was found to alleviate depressive symptoms, insomnia, dizziness and hot flushes. Attempts were made to investigate the possible influence of serum estrogen levels on the observed effects; however, no conclusive results were obtained [207]. Another study conducted in Iran with 66 postmenopausal women aged 45 to 55 years examined the effects of consuming 100 g of yogurt containing *Lactobacillus bulgaricus*, *Streptococcus thermophilus*, *Bifidobacterium lactis* and *Lactobacillus acidophilus* daily for six weeks. The results showed a reduction in anxiety and stress levels; however, no significant effects were found on depressive symptoms or sleep quality [208].

Although clearly defined phases such as pregnancy, postpartum, and menopause are associated with an increased susceptibility to mental disorders in women, symptoms of anxiety and depression can also occur outside these periods. In this context, a study conducted in Iran examined women who had been diagnosed with major depressive disorder who were undergoing antidepressant treatment and were attending a psychiatric clinic. The participants were given an oral capsule containing a combination of *Lactobacillus acidophilus, Bifidobacterium bifidum, Lactobacillus reuteri* and *Lactobacillus fermentum* daily for two months. The results showed a reduction in depression scores as well as an improvement in sexual function and satisfaction [209]. Furthermore, pathologies related to hormonal imbalances have also been implicated in the development of anxiety and depression symptoms in women. One of the most commonly reported disorders in this context is PCOS. This disorder is characterized by excessive androgen levels, ovulatory dysfunction, a high number of preantral follicles in the ovaries and a general state of inflammation. Clinically, it often manifests as insulin resistance or hyperinsulinemia [210].

Given the association between PCOS and systemic inflammation, as well as comorbidity with mental health disorders such as anxiety and depression, the potential effects of probiotics on the emotional well-being of affected women have also been explored, although they have been less studied. For example, a study by [211] examined women aged 18 to 40 years with PCOS who received a co-administration of vitamin D and a probiotic supplement—comprising *Lactobacillus acidophilus, Bifidobacterium bifidum, Lactobacillus reuteri* and *Lactobacillus fermentum*—simultaneously for 12 weeks. The intervention led to a significant improvement in anxiety and depression symptoms. These effects were accompanied by a reduction in serum CRP levels, suggesting a regulatory role of probiotics in the underlying inflammatory processes that may influence mood in this population. Table 2 highlights the current state of clinical trials assessing probiotic interventions in women with emotional or mood-related outcomes.

In conclusion, these clinical findings in menopausal women may be supported by the molecular mechanisms shown in Figure 1 and Figure 2, where reduced neuroinflammation and hormonal modulation appear to be key to improving symptoms of anxiety and depression. Additionally, probiotics could represent a promising therapeutic alternative for treating anxiety and depressive symptoms in women, especially during the reproductive phase, which is characterized by hormonal fluctuations that increase susceptibility to mental disorders. However, the available evidence remains limited and heterogeneous, with significant gaps regarding the standardization of strains, dosages and treatment durations, as well as the role of hormonal variables as potential mediators. Therefore, a more comprehensive and interdisciplinary research approach is needed to better define the modulating potential of probiotics on women’s mental health.

## 8. Therapeutic and Preventive Applications

The use of probiotics as adjuvants has recently gained interest [212]. Mood disorders due to hormonal imbalance and the role of probiotics in modulating the gut–brain axis, especially hormonal changes such as estrogen decline [12]. Probiotics can influence mood by synthesizing neurotransmitters such as serotonin and GABA. In addition, they can reduce symptoms of anxiety and depression associated with hormonal imbalances by reducing cortisol levels [213]. On the other hand, clinical and preclinical studies have shown that probiotic formulations reduce cortisol levels in urine and saliva and also improve psychological well-being [16,17,214]. It is known that hormone fluctuations and cortisol are directly linked to the development of mood disorders and that probiotics can restore hormone and neurochemical concentrations and thus improve mood and emotional health [215].

In addition, preclinical studies have provided important mechanistic insights into the protective effect of probiotics. In ovariectomized mouse models simulating postmenopausal estrogen deficiency, probiotic supplementation showed a protective effect against menopausal symptoms, including mood disorders [216]. These results suggest that probiotics may act as endocrine modulators and influence the synthesis and metabolism of steroid hormones via the intestinal estrobolome.

In certain populations, such as menopausal women, probiotics have shown unique abilities to influence serum estrogen levels. A controlled clinical trial conducted in 2024 showed that supplementation with a probiotic formula with β-glucuronidase activity was effective in modulating serum estrogen levels in healthy peri- and postmenopausal women [217]. This hormonal modulation is particularly important as estrogen loss is associated with a variety of symptoms that can affect quality of life, including neuropsychiatric manifestations such as anxiety and depression.

It has been suggested that taking probiotics increases estradiol levels in pre-menopausal women, while testosterone levels are lower in menopausal women. It is thought that probiotics may modulate levels of hormones related to mood regulation. However, further research is needed to support probiotics as a promising complementary therapy to improve mood problems [218,219].

There is evidence for the effect of probiotics on cognitive function, anxiety, depression and memory impairment due to microbiota dysbiosis. However, the clinical evidence for the use of probiotics as adjuvants in the treatment of mood disorders due to hormonal imbalance is unclear [12]. Estrogens are known to stimulate neuronal growth and play a fundamental role in memory and mood regulation; it has been reported that dysbiosis of the gut microbiota can affect estrogen metabolism and thus exacerbate symptoms of mood disorders under physiological conditions of hormonal imbalance. Estrogen levels are regulated by the “estrobolome” of the gut microbiota, which contains bacterial genes encoding enzymes such as β-glucuronidases and β-glucosidases [65]. In this sense, the effect of probiotics on modulating the microbiota could promote hormonal balance and emotional states [12].

A controlled clinical trial investigated whether short-term, high-dose probiotic supplementation can reduce depressive symptoms in patients diagnosed with major depressive disorder, in addition to changes in gut microbiota composition and brain activity. Probiotics were shown to reduce depressive symptoms, maintain the diversity of the gut microbiota and alter the activity of brain structures responsible for emotional and motivational states. In this sense, microbiota-based therapies could be a complementary strategy in the treatment of depression [220]. However, the development of specific and effective probiotics for the symptoms of depression, as well as research into their mechanisms regarding the gut–brain axis, is one of the greatest challenges for improving patients’ quality of life [221].

In terms of prevention, there is evidence that probiotics can prevent the development of mood disorders in at-risk groups. A five-week randomized clinical trial in perimenopausal and postmenopausal women showed that probiotics affect Follicle-Stimulating Hormone (FSH) levels in perimenopausal women while providing a non-invasive strategy to influence hormonal homeostasis [222]. This preventive approach is particularly important considering that hormonal fluctuations during perimenopause represent a significant risk factor for the development of mood disorders.

The clinical implementation of these probiotic interventions requires specific considerations regarding dosage, duration of treatment and strain selection. Recent meta-analyzes suggest that therapeutic effects are strain-specific and dose-dependent, with a treatment duration of at least 8–12 weeks showing the most consistent benefit. In addition, the combination of probiotics with other bioactive compounds such as soy isoflavones has shown synergistic effects in modulating the gut microbiome and alleviating metabolic disorders associated with estrogen deficiency [223].

## 9. Challenges and Recommendations for Future Research

While the ovaries and placenta constitute the primary sources of estrogen in premenopausal women, it is essential to recognize that estrogen biosynthesis also occurs in extragonadal tissues, including adipose tissue, adrenal glands, bone, vascular endothelium and the brain itself [224]. This extragonadal aromatization, catalyzed by the enzyme aromatase (CYP19A1), converts androgens to estrogens and represents a significant source of circulating estrogens in males, postmenopausal women and individuals with hypogonadism [225]. Emerging evidence suggests that gut microbiota composition influences systemic inflammation and metabolic signaling pathways that regulate aromatase expression in adipose and other peripheral tissues [226]. Consequently, probiotic-mediated alterations in the gut microbiome may affect estrogen bioavailability in both sexes through multiple mechanisms: (1) the modulation of β-glucuronidase activity affecting estrogen enterohepatic circulation [114], (2) influence on insulin sensitivity and inflammatory cytokines that regulate aromatase expression in adipocytes [227] and (3) potential direct or indirect effects on adrenal steroidogenesis [228]. This broader perspective is particularly relevant when considering the therapeutic potential of probiotics for mood disorders across diverse populations, including males and postmenopausal women, where extragonadal estrogen synthesis predominates.

Despite promising progress in understanding the estrogenic effect of probiotics on anxiety and depression, there are important limitations and challenges that need to be addressed to consolidate this field of research. First, there remains a significant lack of consensus on the actual magnitude of the estrogenic effect of probiotics, mainly due to the methodological heterogeneity of available studies. The variability of treatment approaches and clinical presentations limits the comparability and generalizability of results and underscores the need for greater standardization of research protocols. In addition, individual response to probiotics is highly variable and is influenced by various factors, such as baseline microbiota composition, diet, lifestyle, genetic factors and the individual’s prior hormonal status. The microorganisms contained in probiotics colonize the gut only temporarily and this varies greatly from person to person, making it difficult to predict therapeutic response and determine the optimal dose. This variability becomes even more complex when considering that estrogenic effects may manifest differentially depending on the predominant source of hormonal synthesis (gonadal versus extragonadal) in each specific population, which has not been sufficiently characterized in the current literature. Finally, there is an urgent need for more controlled clinical trials and longitudinal studies specifically investigating the estrogenic effects of probiotics in populations with mood disorders, stratified by sex, age and reproductive status. The variability in the efficacy of strains underscores the need for standardization and targeted studies that allow for the development of robust evidence-based clinical recommendations for the use of probiotics as hormone modulators in the context of mental health, considering both gonadal and extragonadal mechanisms of estrogenic production.

## 10. Conclusions

In conclusion, recent studies report that *Lactobacillus* and *Bifidobacterium* can modulate estrogen levels via the estrobolome and affect brain function and mood through mechanisms that include neurotransmitter regulation, HPA axis modulation, MGB axis modulation and neurotransmitter regulation. Preclinical studies report improvements in anxiety- and depression-related behaviors following chronic probiotic supplementation, while clinical studies suggest benefits in perimenopausal and postmenopausal women due to the ability of probiotics to influence estrogen metabolism through β-glucoronidase. However, reports on the effect of probiotics on mental health and estrogen regulation should be interpreted with caution due to methodological limitations such as heterogeneity in study designs, individual variability in response to probiotic treatment and a lack of standardization in preclinical and clinical interventions, presenting significant challenges that require priority attention. Future research should include more comprehensive clinical trials with homogeneous populations and standardized protocols that allow causal links to be established between probiotic supplementation, estrogen changes and mood improvements. In addition, the focus should be on identifying predictive biomarkers of response and developing personalized approaches that account for individual variability in microbiota composition and hormone profile. Overcoming these deficits could make probiotics an important part of the global effort to mitigate symptoms of anxiety and depression in women with hormonal fluctuations.

## Figures and Tables

**Figure 1 ijms-26-09948-f001:**
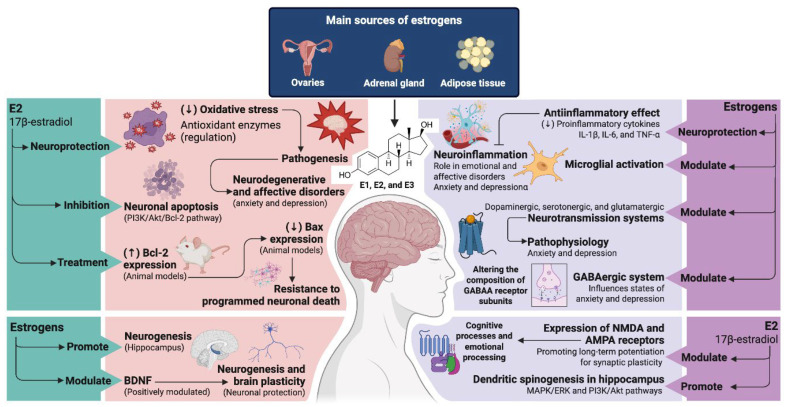
Neuroprotective mechanisms of estrogens: antioxidant, anti-apoptotic, anti-inflammatory, and neurotransmission modulation pathways. The figure illustrates the main neuroprotective mechanisms of estrogens in the brain. Estrogens exert antioxidant effects by upregulating enzymes such as superoxide dismutase and glutathione peroxidase, inhibit apoptosis through PI3K/Akt/Bcl-2 signaling, and promote hippocampal neurogenesis by modulating BDNF. Furthermore, they reduce neuroinflammation by decreasing proinflammatory cytokine production and microglial activation, and regulate 5-HT, DA, glutamatergic, and GABAergic neurotransmission systems. These actions, together, improve synaptic plasticity, dendritic spinogenesis, and cognitive–emotional regulation. Figure created with BioRender (https://www.biorender.com, accessed on 20 August 2025).

**Figure 2 ijms-26-09948-f002:**
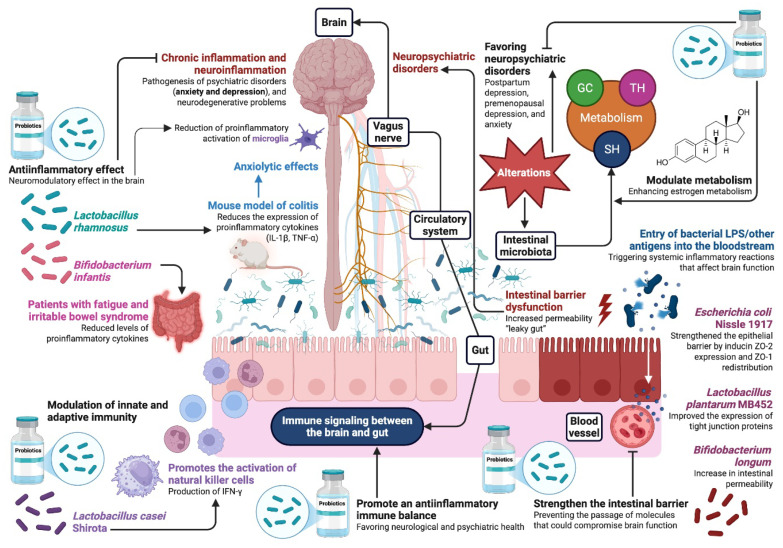
Potential mechanisms of action of probiotics. The figure shows the potential mechanisms by which probiotics exert neuroprotective and neuromodulatory effects through the MGB axis. Probiotics reduce systemic and neuroinflammatory inflammation by downregulating proinflammatory cytokines and microglial activation, enhance SCFAs production, and protect the integrity of the BBB. They strengthen gut barrier function by upregulating tight junction proteins, modulate innate and adaptive immunity, and influence the metabolism of neuroactive compounds such as estrogens through the estrobolome. These mechanisms together contribute to improved emotional regulation, reduced depressive- and anxiety-like behaviors, and potential therapeutic benefits in estrogen-related mood disorders. Figure created with BioRender (https://www.biorender.com, accessed on 25 August 2025).

**Table 1 ijms-26-09948-t001:** Preclinical evidence of the effect of probiotics on anxiety and depression (the up arrow indicates increase and down arrow indicates decrease).

Probiotic	Experimental Subject	Dosage and Treatment Time	Experimental Model	Effect on Anxiety and Depression	Mechanism	Reference
*Lacticaseibacillus rhamnosus* LB1.5	Male mice C57BL/6.	3.1 × 10^8^ CFU/mL three times a week for 13 weeks.	Mice on a high-fat diet Light–dark test.	↓ Anxiety (↑ distance and time in illuminated compartment in light–dark test).	↓ IL-6, neuroprotection, reduction in neuroinflammation.	[173].
*Limosilactobacillus reuteri*	Male mice C57BL/6.	2.0 × 10^9^ CFU/0.1 mL daily for 9 weeks.	Mice with sleep deprivation and water restriction Elevated plus-maze Y maze test.	↓ Anxiety (↑ distance and time in open arms in elevated plus-maze and Y maze test).	↓ Cortisone and CRH; positive modulation of the HPA axis; ↓ visceral hypersensitivity.	[60].
Mixture: *Bifidobacterium*, *Lactobacillus acidophilus*, *L. rhamnosus*, *Bifodan* A/S	Male mice C57BL/6N.	1 × 10^9^ CFU/mL, 200 µL/daily for 30 days.	Mice with chronic exposure to ethanol Elevated plus-maze Open field test.	↓ Anxiety (reversal of effects on elevated plus-maze and open field test).	↓ NLRP3, NF-κB, IL-1β in hippocampus; reduction in neuroinflammation via extracellular vesicles.	[174].
*Bacillus coagulans* Unique IS-2	Female and male Sprague Dawley rats.	2 × 10^9^ CFU daily for 6 weeks.	Rats with maternal separation + CUMS. Elevated plus-maze Forced swim test Sugar water consumption test.	↓ Anxiety and depression (elevated plus-maze; forced swim test; sugar water consumption test). Effects independent of sex.	↑ Hippocampal BDNF; ↓ TNF-α, CRP, IL-1β; ↑ L-tryptophan; sex-dependent effects on monoamines: ♂ ↑ DA reversal; ♀ ↑↑ DA and 5-HT (double effect).	[175].
Mixture: *Bifidobacterium breve* (25%), *Lactobacillus plantarum* (25%), *L. paracasei* (25%), *L. helveticus* (25%)	Male mice C57BL/6J.	2, 4 and 8 × 10^9^ CFU daily for 10 days.	Mice with chronic unpredictable stress (CUS) Tail suspension test Forced swim test Open field test Sucrose preference test.	↓ Depression (↑ sucrose preference; ↓ tail suspension immobility and forced swimming).	↓ IL-1β, IL-6, TNF-α; ↑ IL-4, IL-10, Ym-1; ↓ nitroxidative stress (NO, MDA); ↑ GSH; ↑ BDNF in hippocampus and medial prefrontal cortex.	[176].
Mixture of probiotics	Male Sprague Dawley rats.	Unspecified dose.	Rats subjected to CUMS Forced swim test Open field test.	↓ Depression (reversal of effects in forced swimming and open field test).	↑ Serotonin and norepinephrine in the hippocampus; ↓ ACTH and CORT in plasma; ↑ *Lactobacillus* and *Bifidobacterium*; ↓ *E. coli* and *E. faecalis*.	[180].
*Bifidobacterium longum* and *L. rhamnosus*	Male Wistar rats.	1 × 10^9^ CFU per 100 g weight for 4 weeks.	Rats subjected to CUMS Forced swim test Sucrose preference test.	↓ Depression (better performance in forced swimming; ↑ sucrose preference).	↓ 5-HT in the colon; ↑ 5-HT in the hippocampus and prefrontal cortex (redistribution associated with antidepressant effect).	[181].
*Lactobacillus rhamnosus* JB-1	Male mice BALB/c.	5 × 10^9^ per ml for 28 days.	Mice subjected to Open field test Stress-induced hyperthermia Elevated plus maze Fear conditioning Forced swim test.	↓ Anxiety and depression (elevated plus-maze; forced swim test.	↓ Corticosterone; modulation of central GABA receptors via the vagus nerve.	[15].
*Lactobacillus rhamnosus* JB-1	Male Wistar rats	0.2 mL of bacteria suspension (~1.7 × 10^9^ CFU) daily for 4 weeks.	Rats subjected to CUMS Elevated plus maze.	Restoration of neurochemical brain balance.	↑ GSH, glutamate and norepinephrine transporters.	[182].
*Lactobacillus casei*	Male Sprague Dawley rats	8 × 10^8^ CFU kg^−1^ day^−1^ for 3 weeks.	Rats were treated with CUMS once a day and lasted for 7 weeks Open field test Forced swim test Sucrose preference test.	↓ Depression.	↑ NE, 5-HT, DA, BDNF, and BDNF-TrkB signaling pathway.	[183].
*Lactobacillus paracasei* HT6	Pregnant Wistar rats.	1 × 10^9^ CFU daily for 14 days, from PND-2 to PND-16.	Maternal separation model (early stress) Open field test Tail suspension test.	↓ Anxiety and depression.	Normalization of ACTH, CORT, GR, 5-HT, NE, DA; regulation of microRNA124a/132 and glutamate receptors.	[184].
*Prevotella histicola*	Female C57BL/6 Mice.	0.2 mL of a *P. histicola* suspension with a concentration of 1 × 10^9^ CFU/mL for 4 weeks.	Ovariectomized mice (estradiol deficiency) Open field test Elevated plus-maze Forced swim test Tail suspension test.	↓ Anxiety (open field, raised arms maze) and depression (tail suspension test and forced swim test).	Reversal of neuronal damage and inflammation in the CA3 region of the hippocampus; ↑↑ serotonergic, dopaminergic, and BDNF receptors (promoting neuronal proliferation); reversal of ovariectomy-induced dysbiosis.	[186].
Mixture: *L. casei*, *L. acidophilus*, *L. rhamnosus*, *L. bulgaricus*, *Bifidobacterium breve*, *B. infantis*, *Streptococcus thermophilus* + FOS	Male Swiss mice	6.25, 12.5, and 25 × 10^6^ CFU for 7–14 days.	Swiss mice, depression model Marble-burying test Forced swim test.	Potentiation of antidepressant effects: Venlafaxine (SNRI): synergistic effect from day 7 in FST and MBT; Doxepin (TCA): synergistic effect from day 7 in FST; Fluvoxamine (SSRI): synergistic effect only after 14 days in FST	Modulation of serotonergic (Marble burying test) and noradrenergic (forced swim test) systems; restoration of monoamine levels in brain regions ↑ Tryptophan attenuation of proinflammatory response; mediated by vagus nerve; more potent synergistic effect with SNRI (venlafaxine)	[187].
*Lactobacillus rhamnosus*	Male mice (C57BL/6 × BALB/c).	1 × 10^9^ CFU daily for 30 days.	Mice treated with sertraline (20 mg/kg)	Reversal of sertraline side effects: Prevention of ↓ body weight; Prevention of ↓ seminal vesicle weight; ↓ number of embryonic resorptions; improvement in exploratory profile; ↓ sertraline-induced anxiety	Protection against adverse effects: ↑ sperm quality; improved natural fertility; regulation of testosterone levels (spermatogenesis); modulation of intestinal microbiota; first demonstration of probiotic alleviating side effects of SSRIs	[188].

**Table 2 ijms-26-09948-t002:** Summary of clinical trials assessing probiotic interventions in women with emotional or mood-related outcomes, including study status and biomarker findings.

Reference	Population (Age, Condition)	Intervention/Probiotic	Duration	Outcomes Evaluated	Main Findings	Biomarker Assessment	Study Status
[194]	65 women (18–40 years	*B. longum* subsp. *longum* OLP-01, *L. plantarum* PL-02, *L. lactis* LY-66	10 weeks	Premenstrual symptoms: emotional (anger, depression, crying, anxiety), fatigue, social limitations	Significant improvement in overall premenstrual symptoms, particularly emotional, fatigue, and social domains vs. placebo	Not assessed	Completed
[195]	Young women, 20–35 years	*L. gasseri* CP2305 tablets, daily	183 days	Anxiety, depression, fatigue	Reduction in anxiety, depression, and fatigue; improvement	Estrogen and progesterone increased during luteal phase	Completed
[200]	Pregnant women, 36 weeks gestation	*L. rhamnosus GG* + *B. lactis* BB12, capsules, daily	36 weeks	Depression, anxiety, functional health, general well-being	No significant improvements observed, possible insufficient dose and floor effect	Not assessed	Completed
[201]	Pregnant women, third trimester (≥21 years)	*Bifidobacterium longum* NCC3001	Third trimester of pregnancy	Anxiety, depression symptoms	No significant effects on anxiety or depression; evidence limited and inconclusive	Not assessed	Completed
[204]	Healthy pregnant women, 18–50 years	*Limosilactobacillus reuteri* PBS072 + *Bifidobacterium breve* BB077	90 days	Anxiety and depression symptoms	Reduction in anxiety and depression	Not assessed	Completed
[207]	Japanese premenopausal women, 40–60 years	*Lactobacillus gasseri* CP2305 tablets, once daily	Six menstrual cycles	Depressive symptoms, insomnia, dizziness, hot flushes	Alleviation of depressive symptoms, insomnia, dizziness, and hot flushes	Serum estrogen measured, no conclusive results	Completed
[208]	Postmenopausal women, 45–55 years	100 g yogurt containing *L. bulgaricus*, *S. thermophilus*, *B. lactis*, *L*. *acidophilus*, daily	6 weeks	Anxiety, stress, depressive symptoms, sleep quality	Reduction in anxiety and stress; no significant effects on depressive symptoms or sleep quality	Not assessed	Completed
[209]	Women with major depressive disorder, on antidepressant treatment	Oral capsules containing *L. acidophilus*, *B. bifidum*, *L. reuteri*, *L. fermentum*, daily	2 months	Depression scores, sexual function, sexual satisfaction	Reduction in depression scores; improvement in sexual function and satisfaction	Not assessed	Completed
[211]	Women with PCOS, 18–40 years	Co-administration of vitamin D + probiotic supplements (*L*. *acidophilus*, *B. bifidum*, *L. reuteri*, *L. fermentum*), daily	12 weeks	Anxiety and depression symptoms	Significant improvement in anxiety and depression	CRP levels reduced	Completed

## Data Availability

Not applicable.

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
