# Peer review of "Estrogenic Effect of Probiotics on Anxiety and Depression: A Narrative Review"

_ijms, 2025, doi:10.3390/ijms26209948_

Round 1
Reviewer 1 Report
Comments and Suggestions for Authors
This review provides a timely and compelling synthesis on the role of probiotics in modulating estrogen-mediated pathways to improve anxiety and depression, particularly in women. The regulation of mood by estrogen through the gut-brain axis and the HPA axis, in conjunction with the health of the gut microbiota, is a rational and effective approach to psychological stability. The integration of preclinical and clinical evidence strengthens the narrative. The review also has good-quality and informative figures.
Key Suggestions for Further Improvement:
-
Clarify Methodology: Specify the review as "systematic narrative" or consider including a PRISMA-style flow diagram (or similar approach) to enhance reproducibility.
-
Strengthen Clinical Linkages: Explicitly tie clinical findings (e.g., in menopausal women) back to mechanistic pathways (e.g., neuroinflammation reduction, Fig. 1/2), if possible.
- Consider briefly referencing if appropriate: 1) Sun Y, Ju P, Xue T, Ali U, Cui D, Chen J. Alteration of faecal microbiota balance related to long-term deep meditation. General Psychiatry. 2023;36:e100893. https://doi.org/10.1136/gpsych-2022-100893 , which provide empirical support showing that meditation alters gut microbiota in ways parallel to your probiotic findings, reinforcing the broader mechanistic link between microbial modulation and mental health (for example in anxiety and mood management). 2) Liu J, Xu K, Wu T, Yao L, Nguyen TT, Jeste D, et al. Deciphering the ‘gut–brain axis’ through microbiome diversity. General Psychiatry. 2023;36:e101090. https://doi.org/10.1136/gpsych-2023-101090, which findings help support the existence of the microbiome–gut–brain axis.
The manuscript is comprehensible but requires moderate editing to correct grammatical errors, improve phrasing, and adhere to the conventions of academic English. This polishing is necessary for publication in a leading international journal.
Author Response
Dear Reviewer, we appreciate your comments on our manuscript; they have undoubtedly improved its quality.
Comments 1: [Clarify Methodology: Specify the review as "systematic narrative" or consider including a PRISMA-style flow diagram (or similar approach) to enhance reproducibility].
Response 1: [Dear reviewer, thank you for your suggestion. We are aware of the potential limitations of the methods report in a narrative review and the importance of your suggestion to include additional elements from PRISMA. However, we have assumed that this review is a narrative review in which some items requested by PRISMA have not been included from the outset. We are concerned about the inclusion of PRISMA elements as we may omit some information and confuse potential readers as this narrative review may contain a mixture of methods. We respectfully request to the reviewer to consider this possibility and permit us to omit this suggestion from our report. To improve our methods report, we have added some clarifications that have been included in the methods and highlighted in red].
Comments 2: [Strengthen Clinical Linkages: Explicitly tie clinical findings (e.g., in menopausal women) back to mechanistic pathways (e.g., neuroinflammation reduction, Fig. 1/2), if possible].
Response 2: [Dear Reviewer, we appreciate your valuable comment. Lines 837–840 link the clinical findings to the possible mechanisms of action involved; the text has been highlighted in red].
Comments 3: [Consider briefly referencing if appropriate: 1) Sun Y, Ju P, Xue T, Ali U, Cui D, Chen J. Alteration of faecal microbiota balance related to long-term deep meditation. General Psychiatry. 2023;36:e100893. https://doi.org/10.1136/gpsych-2022-100893 , which provide empirical support showing that meditation alters gut microbiota in ways parallel to your probiotic findings, reinforcing the broader mechanistic link between microbial modulation and mental health (for example in anxiety and mood management). 2) Liu J, Xu K, Wu T, Yao L, Nguyen TT, Jeste D, et al. Deciphering the ‘gut–brain axis’ through microbiome diversity. General Psychiatry. 2023;36:e101090. https://doi.org/10.1136/gpsych-2023-101090, which findings help support the existence of the microbiome–gut–brain axis].
Response 3: [
We thank the reviewer for their valuable bibliographic suggestions. We have carefully evaluated both proposed references:
Regarding Sun et al. (2023) on meditation and microbiota:
While we recognize the scientific merit of this study, we have decided not to include it in our review for the following methodological and conceptual reasons:
Topic specificity: Our narrative review focuses specifically on the estrogenic effect of probiotics on anxiety and depression. The inclusion of non-probiotic interventions such as meditation would dilute the central focus of the manuscript and divert attention from the specific mechanism we are analyzing: probiotic-mediated estrogen modulation.
Fundamental mechanistic difference: Although both meditation and probiotics can modulate the gut microbiota, the underlying mechanisms are substantially different. Meditation acts primarily through descending neurophysiological pathways (gut-brain axis), whereas probiotics exert direct ascending effects through microbial colonization and production of specific metabolites. This mechanistic difference makes the comparison less relevant for our purposes.
Lack of estrogenic component: The meditation study does not address the estrogenic hormonal dimension, which is the central and differentiating focus of our review.
Regarding Liu et al. (2023) on the microbiota-gut-brain axis:
This reference, although important, presents similar considerations:
Existing coverage: Our manuscript already includes fundamental and updated references that firmly establish the existence and functioning of the microbiota-gut-brain axis in sections 4, 4.1, 4.2, and 4.3. Adding this reference would be redundant, as it does not provide specific novel information on the estrogenic effect of probiotics.
General vs. specific approach: Liu et al. provides a broad view of the gut-brain axis, while our review requires specific evidence linking probiotics, estrogen modulation, and mood disorders. The references already included address this specificity more directly.
Narrative coherence: Maintaining the focus on studies that specifically address probiotics and/or estrogenic mechanisms ensures the coherence and clarity of the scientific argument developed throughout the manuscript.
We believe that including these references, although scientifically valid, would compromise the specificity and thematic coherence of our narrative review without adding direct evidence on the specific estrogenic effect of probiotics on anxiety and depression. Our current bibliographic selection provides the empirical and theoretical support necessary for the arguments presented, while maintaining focus on the stated objective of the manuscript.
We once again thank the reviewer for his attention and interest in strengthening our work].
Reviewer 2 Report
Comments and Suggestions for Authors
The author has discussed the estrogenic effects of prebiotics on anxiety and depression in a detailed and engaging manner. However, there are a few minor points that could enhance the clarity and structure of the article:
1. The author has presented both clinical and preclinical studies; however, it is recommended to present this information in a tabular format for better clarity and ease of understanding. A concise table summarizing the role of the discussed prebiotics, their effects on estrogen, and how they positively regulate anxiety and depression would be highly beneficial.
2. Clinical trial-related studies should be summarized in a separate table, highlighting their current status (e.g., completed, ongoing, or pending). This will provide a clearer overview of the progress in this field.
3. The author’s perspective on the estrogenic effects of prebiotics could be compiled in a paragraph to clarify whether these effects are specifically related to the ovaries/placenta, or if prebiotics also influence estrogen levels in males through the adrenal glands and fat tissues.
Author Response
Dear Reviewer, we appreciate your comments on our manuscript; they have undoubtedly improved its quality.
Comments 1: [The author has presented both clinical and preclinical studies; however, it is recommended to present this information in a tabular format for better clarity and ease of understanding. A concise table summarizing the role of the discussed prebiotics, their effects on estrogen, and how they positively regulate anxiety and depression would be highly beneficial].
Response 1[Dear Reviewer, we appreciate your recommendation. We have included a table in Section 6, Preclinical Evidence of Probiotic Use, describing the effects of probiotics on estrogen receptors and how this modulation regulates anxiety and depression].
Comments 2: [Clinical trial-related studies should be summarized in a separate table, highlighting their current status (e.g., completed, ongoing, or pending). This will provide a clearer overview of the progress in this field].
Response 2: [Dear Reviewer, we appreciate your recommendation. We have included a table in Section 7, Clinical Evidence of the Effects of Probiotics on Depressive and Anxiety Symptoms in Women, highlighting their current status (e.g., completed, ongoing, or pending)].
Comments 3: [The author’s perspective on the estrogenic effects of prebiotics could be compiled in a paragraph to clarify whether these effects are specifically related to the ovaries/placenta, or if prebiotics also influence estrogen levels in males through the adrenal glands and fat tissues].
Response 3: [
We sincerely appreciate the reviewer’s observation regarding the need to clarify the scope of the estrogenic effects of probiotics. We recognize that our current manuscript may have inadvertently focused primarily on gonadal estrogen production without adequately addressing the broader context of extragonadal estrogen synthesis.
In response to this valuable feedback, we have added two new paragraphs, highlighted in red, to the requested section. This addition clarifies that while ovarian and placental estrogen production is the primary source in premenopausal women, probiotics may also influence estrogen levels by modulating extragonadal synthesis in adipose tissue, adrenal glands, and other peripheral tissues. This mechanism is particularly relevant in men, postmenopausal women, and individuals with compromised gonadal function, where extragonadal aromatization becomes the predominant source of circulating estrogen.
We believe this clarification strengthens the manuscript by providing a more comprehensive understanding of how probiotics may exert estrogenic effects in different populations and physiological contexts, thereby improving the translational relevance of our review].